# Aβ43-producing PS1 FAD mutants cause altered substrate interactions and respond to γ-secretase modulation

Johannes Trambauer[1] (iD), Rosa María Rodríguez Sarmiento[2], Akio Fukumori[3,4], Regina Feederle[5,6] (iD), Karlheinz Baumann[2] & Harald Steiner[1,6,*] (iD)

## Abstract

Abnormal generation of neurotoxic amyloid-β peptide (Aβ) 42/43 species due to mutations in the catalytic presenilin 1 (PS1) subunit of γ-secretase is the major cause of familial Alzheimer's disease (FAD). Deeper mechanistic insight on the generation of Aβ43 is still lacking, and it is unclear whether γ-secretase modulators (GSMs) can reduce the levels of this Aβ species. By comparing several types of Aβ43-generating FAD mutants, we observe that very high levels of Aβ43 are often produced when presenilin function is severely impaired. Altered interactions of C99, the precursor of Aβ, are found for all mutants and are independent of their particular effect on Aβ production. Furthermore, unlike previously described GSMs, the novel compound RO7019009 can effectively lower Aβ43 production of all mutants. Finally, substrate-binding competition experiments suggest that RO7019009 acts mechanistically after initial C99 binding. We conclude that altered C99 interactions are a common feature of diverse types of PS1 FAD mutants and that also patients with Aβ43-generating FAD mutations could in principle be treated by GSMs.

**Keywords** Aβ 43; amyloid β-peptide; familial Alzheimer's disease; presenilin; γ-secretase modulator

**Subject Categories** Molecular Biology of Disease; Neuroscience

## Introduction

Accumulation and deposition of amyloid-β peptide (Aβ) species is a major pathological hallmark of Alzheimer's disease (AD) [1]. The various Aβ species, 37–43 amino acids in length, are generated from the C99 fragment of the amyloid precursor protein (APP) by presenilins 1 and 2 (PS1 and PS2), the catalytic subunits of the intramembrane-cleaving protease γ-secretase [2]. Following an initial cleavage of the C99 transmembrane domain at the ε-site, which releases the APP intracellular domain (AICD) and gives rise to Aβ49 or Aβ48, further stepwise carboxy-terminal cleavages occur [2]. Aβ49 is sequentially cleaved in a major product line to Aβ46, Aβ43, and Aβ40, the main Aβ species, as well as small amounts of Aβ37 (Aβ40 product line), while Aβ48 undergoes sequential cleavages in an alternative product line to Aβ45, Aβ42, and Aβ38 (Aβ42 product line) [3]. The longer Aβ42 and Aβ43 species are highly aggregation-prone and neurotoxic and considered as the primary trigger of the disease [4]. Since lowering of Aβ should be protective against AD pathogenesis, γ-secretase is a target for AD therapy. Small molecules targeting the enzyme that hold great potential to be beneficial in AD are γ-secretase modulators (GSMs) [5]. GSMs alter the cleavage of C99 toward the production of non-toxic shorter Aβ species [6] and maintain the cleavage of physiologically important γ-secretase substrates [7,8]. These compounds would therefore offer a preferred treatment strategy over a broad inhibition of substrate cleavage, which is considered a main reason among others, such as pseudo-inhibition [9], for the failure of γ-secretase inhibitors in clinical trials [10].

Mutations in PS1 represent the major cause of early-onset familial AD (FAD). They lead to the generation of abnormal relative amounts of the longer Aβ forms as a result of a reduced carboxy-terminal processivity [11,12]. In support of their pathogenicity, changes in the ratios of Aβ42 to Aβ40 correlate with the clinical age of onset of the particular PS1 FAD mutant [13]. While it has been well established that Aβ42/Aβ40 ratios are increased for PS1 and PS2 FAD mutants [14,15], in a few cases such as the PS1 R278I or PS1 L435F mutants increased levels of Aβ43 rather than Aβ42 are observed [16–18]. For these mutants, the processivity changes appear also to be associated with a stronger loss of function in the ε-site cleavage of C99 [16,17,19]. However, since deficiencies in the initial ε-site cleavage are largely compensated by the unaffected wt PS1 and PS2 alleles in human FAD brain, while pathogenic Aβ ratios

1 Biomedical Center (BMC), Metabolic Biochemistry, Ludwig-Maximilians-University, Munich, Germany
2 Roche Pharma Research and Early Development, Roche Innovation Center Basel, F. Hoffmann-La Roche Ltd., Basel, Switzerland
3 Department of Aging Neurobiology, National Center for Geriatrics and Gerontology, Obu, Japan
4 Department of Mental Health Promotion, Osaka University Graduate School of Medicine, Toyonaka, Japan
5 Institute for Diabetes and Obesity, Monoclonal Antibody Core Facility, Helmholtz Center Munich, German Research Center for Environmental Health, Neuherberg, Germany
6 German Center for Neurodegenerative Diseases (DZNE), Munich, Germany
*Corresponding author. Tel: +4989440046535; E-mail: harald.steiner@med.uni-muenchen.de

persist [20], it is unlikely that FAD mutant γ-secretases show an impact on signaling functions.

PS1 and PS2 normally undergo presenilin endoproteolysis [21] in which they are autoproteolytically cleaved into N- and C-terminal fragments (NTF, CTF) [2]; however, in a few FAD mutants, this step is blocked or impaired. This is also and prominently seen for the PS1 R278I and PS1 L435F mutants [16–19] indicating that their strong deficiency in presenilin endoproteolysis may contribute to their loss in processivity thereby resulting in the formation of longer Aβ43 species.

In an attempt to better understand the molecular characteristics and disease mechanisms of clinical presenilin mutants, we recently observed that two Aβ42-increasing FAD mutants show altered interaction/positioning of the C99 cleavage site domain within the active site region of γ-secretase [22]. In the current study, we wanted to examine whether this may also be a property for the Aβ43-increasing mutants, possibly even more severe, or whether their altered cleavage activities may be unrelated to substrate positioning changes. Since Aβ43 is considered to be the major pathogenic species for these FAD cases, we also investigated whether a GSM could lower the aberrant Aβ43 levels generated by these mutants and whether such a compound would have effects on the interaction with C99.

We found significant changes in the C99 interactions with γ-secretase around the ε-sites for the Aβ43-generating PS1 FAD mutants, showing that altered interactions of this region of C99 with PS1 can also be linked to the aberrant generation of this long Aβ variant. In contrast to other well-characterized GSMs, RO7019009, a novel and potent GSM, could lower Aβ43 for all mutants analyzed. Although RO7019009 could affect to some extent the interaction of C99 with WT PS1 and some of the mutants, there was no clear correlation between these changes and the capability of the GSM to lower Aβ42/43. We conclude that Aβ43 can be targeted by GSMs and that altered substrate interactions are a common characteristic of presenilin FAD mutants associated with the generation of pathogenic Aβ species.

# Results

### Altered substrate positioning by Aβ43-generating PS1 FAD mutants

To investigate whether altered substrate interactions are a molecular characteristic of Aβ43-generating PS1 FAD mutants, we generated HEK293 cells stably overexpressing Swedish (sw) mutant APP (HEK293/sw) in combination with presenilin mutants that produce different amounts of Aβ43. In the first set of mutants, we included strong loss-of-function FAD mutants, which were also deficient in presenilin endoproteolysis. This set contained the previously characterized PS1 R278I and L435F mutants as well as the PS1 V261F mutant, which we identified as additional extreme Aβ43-generating loss-of-function mutant by candidate screening. As a control, we included the uncleavable PS1 M292D mutant, which is known to support APP processing despite being deficient in presenilin endoproteolysis [23]. In a second set, we investigated normally endoproteolysed PS1 mutants. This set included the PS1 L166P and Y256S FAD mutants, which generate high amounts of Aβ42 as the prevailing pathogenic Aβ species besides Aβ43 [24] as well as the synthetic

G382A mutant, which generates higher amounts of Aβ43 than Aβ42 [25] and is therefore predicted to cause FAD if it occurred in humans. As shown in Fig 1A and B, the mutants of both categories showed the expected behavior in presenilin endoproteolysis and APP processing as judged from their respective effects on PS1 NTF and CTF as well as total secreted Aβ levels. Tris–Bicine urea SDS–PAGE, mass spectrometry, and ELISA documented the expected profiles of the individual Aβ species generated by the mutants and identified very high relative amounts of Aβ43 for the previously uncharacterized PS1 V261F mutant (Fig 1B and C and Appendix Fig S1).

Next, we asked whether the mutants would show altered substrate interactions with γ-secretase as recently shown for the PS1 A246E and L166P FAD mutants [22]. To this end, we used our previously established photocrosslinking approach by which substrate binding to CHAPSO-solubilized γ-secretase is assessed with recombinant C99 variants engineered to carry the photocrosslinkable non-natural amino acid p-benzoylphenylalanine (Bpa) at any residue of interest [22]. By probing the interactions of the most prominent crosslinking residues that were determined in the previous study for C99 in the major substrate-binding site of γ-secretase, the PS1 NTF, we observed similar interaction profiles for PS1 WT and the uncleaved M292D mutant. The major contact point of C99 was at V44, and additional interactions at the ε-sites T48 and L49 were observed. Being expressed as an uncleaved presenilin, the M292D variant also showed crosslinking at L52, which interacts with PS1 WT primarily in the PS1 CTF [22] (Fig 2A). The FAD mutants were generally interacting with the substrate differently from the WT and M292D controls (Fig 2A). V44 remained the major interaction site of C99 for all mutants. Some of them, however, displayed reduced V44 crosslinking as prominently seen for the R278I mutant (Fig 2A). The strong loss-of-function mutants showed similar alterations at the T48/L49 sites (Fig 2A) as reflected by increased ratios of T48/L49 crosslinking efficiencies (Fig 2B) compared to the corresponding M292D control. In addition, they showed altered interactions with C99 residues C-terminal of the ε-cleavage sites (Fig 2A). Particularly, the V261F and R278I mutants displayed increased crosslinking at K54 relative to the major interaction site V44 (Fig 2C). Consistent with our previous results [22], the L166P mutant showed shifts in substrate crosslinking at the T48 and L49 sites compared to WT (Fig 2A and B). This was also seen for the Y256S mutant, although in opposite direction (Fig 2B). The PS1 G382A mutant showed less pronounced alterations at these sites (Fig 2A and B) but a strong increase in crosslinking at L52 (Fig 2A). Taken together, the mutants showed distinct and individual changes of C99 crosslinking as compared to the respective WT and M292D controls. We conclude that altered substrate positioning is also observed for FAD mutants that generate Aβ43.

### The novel GSM RO7019009 effectively lowers Aβ43 of all Aβ43-generating mutants investigated

Previous studies showed that aberrant Aβ42 generation could be modulated with potent second-generation GSMs for many FAD mutants [26–28], with only a few exceptions such as the PS1 L166P mutant [26,27]. To analyze whether Aβ43 could also be lowered by such GSMs, two compounds of different structural classes were tested on the prototypic uncleaved PS1 R278I mutant and on the normally endoproteolysed PS1 L166P mutant. GSM-1

                                                   

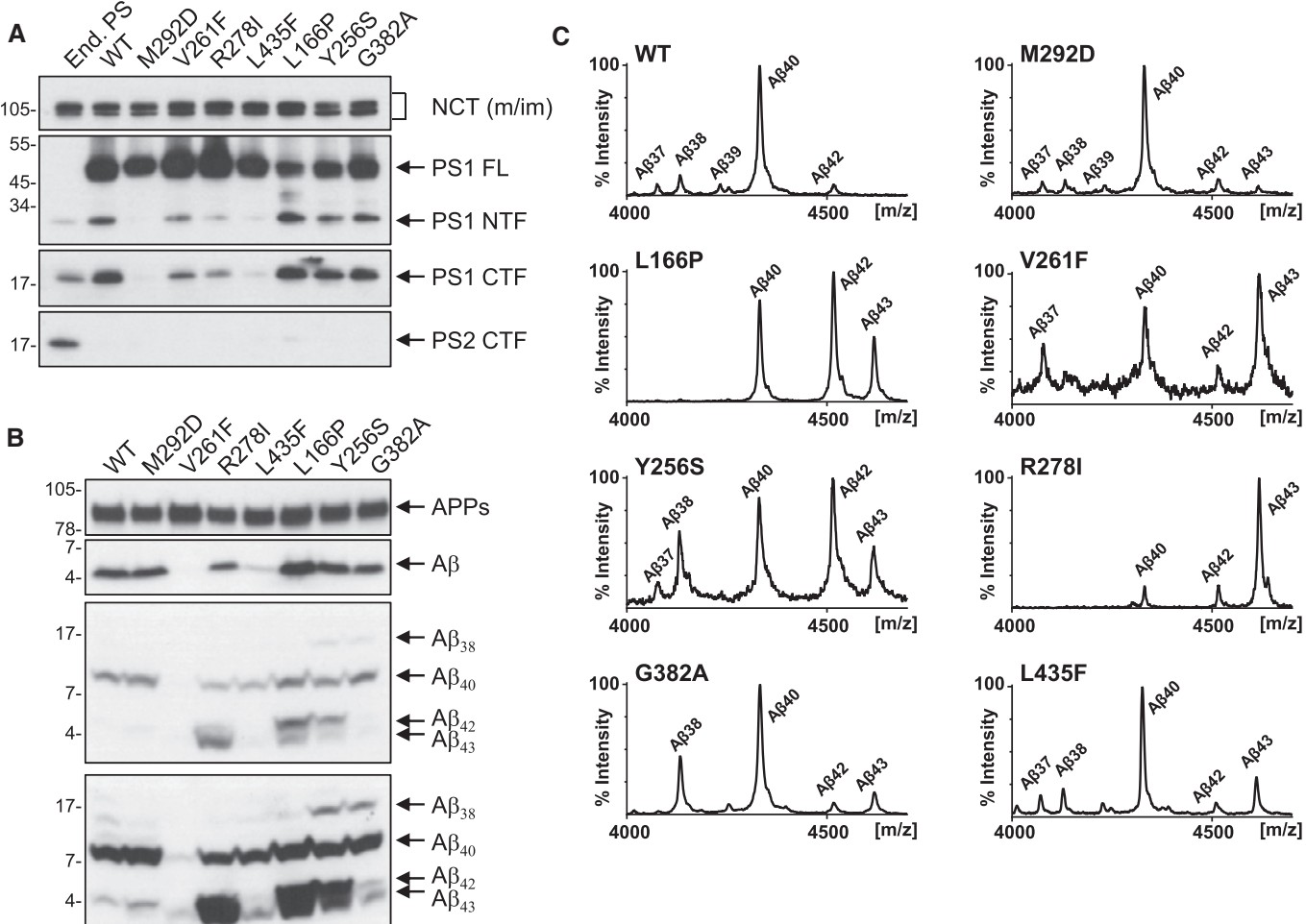

**Figure 1. Expression, endoproteolysis, and APP processing of Aβ43-generating PS1 FAD mutants.**

A  Immunoblot analysis of presenilin expression and endoproteolysis in cell lysates of untransfected HEK293/sw cells endogenously expressing presenilin and single-cell clones overexpressing PS1 WT or the indicated PS1 mutants.

B  Immunoblot analysis of total Aβ secretion and the production of different Aβ species (shown in low and high immunoblot exposures) in conditioned media of cell lines described in (A).

C  Mass spectrometry analysis of Aβ species immunoprecipitated from conditioned media of cell lines described in (A).

Source data are available online for this figure.

[24] (Fig EV1A) was used as a representative of the acidic GSMs and RO-02 [29] (Fig EV1B) as one of the non-acidic GSMs with a bridged aromatic scaffold [5]. As shown in Fig EV1C and D, both modulators weakly reduced Aβ43 production for R278I and only poorly, if at all, for L166P. Aβ42 levels remained unaffected by both GSM-1 and RO-02 treatment. We thus tested the novel compound RO7019009 (Figs 3A and EV2), which shows favorable CNS drug-like properties [30] and has excellent metabolic stability (Table EV1). RO7019009 efficiently reduced the levels of Aβ42 and Aβ40 secreted by HEK293/sw cells in a dose-dependent manner while concomitantly increasing that of Aβ38 (Fig 3B). An $IC_{50}$ of 14 nM for the inhibition of Aβ42 in these cells characterized RO7019009 as one of the most potent GSMs (Fig 3C and D). The $IC_{50}$ (Aβ42) values were even lower for brain-derived human

**Figure 2. Aβ43-generating PS1 FAD mutants show altered C99 interactions.**

A  Immunoblot analysis of C99–PS1 NTF or full-length (FL) crosslinks (CL) of the indicated mutants. Samples that were not UV-irradiated were loaded to control for specificity.

B  Changes in the ratio of crosslink efficiencies at positions T48 and L49 compared to corresponding controls (*n* = 4–5 biological replicates).

C  Changes in the ratio of crosslink efficiencies at positions K54 and V44 compared to corresponding controls (*n* = 4–5 biological replicates).

Data information: Data in (B) and (C) are presented as mean ± SD.
Source data are available online for this figure.

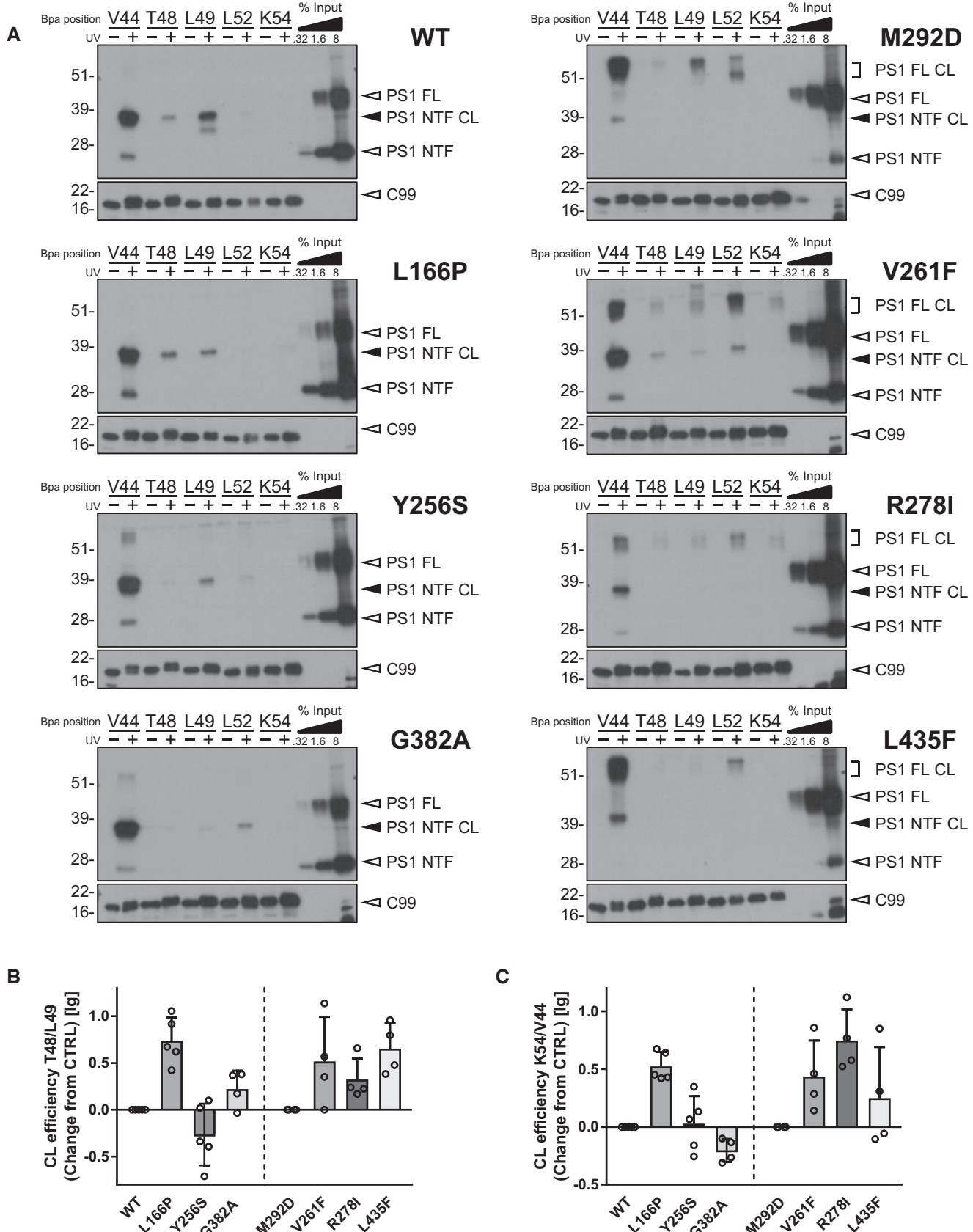

Figure 2.

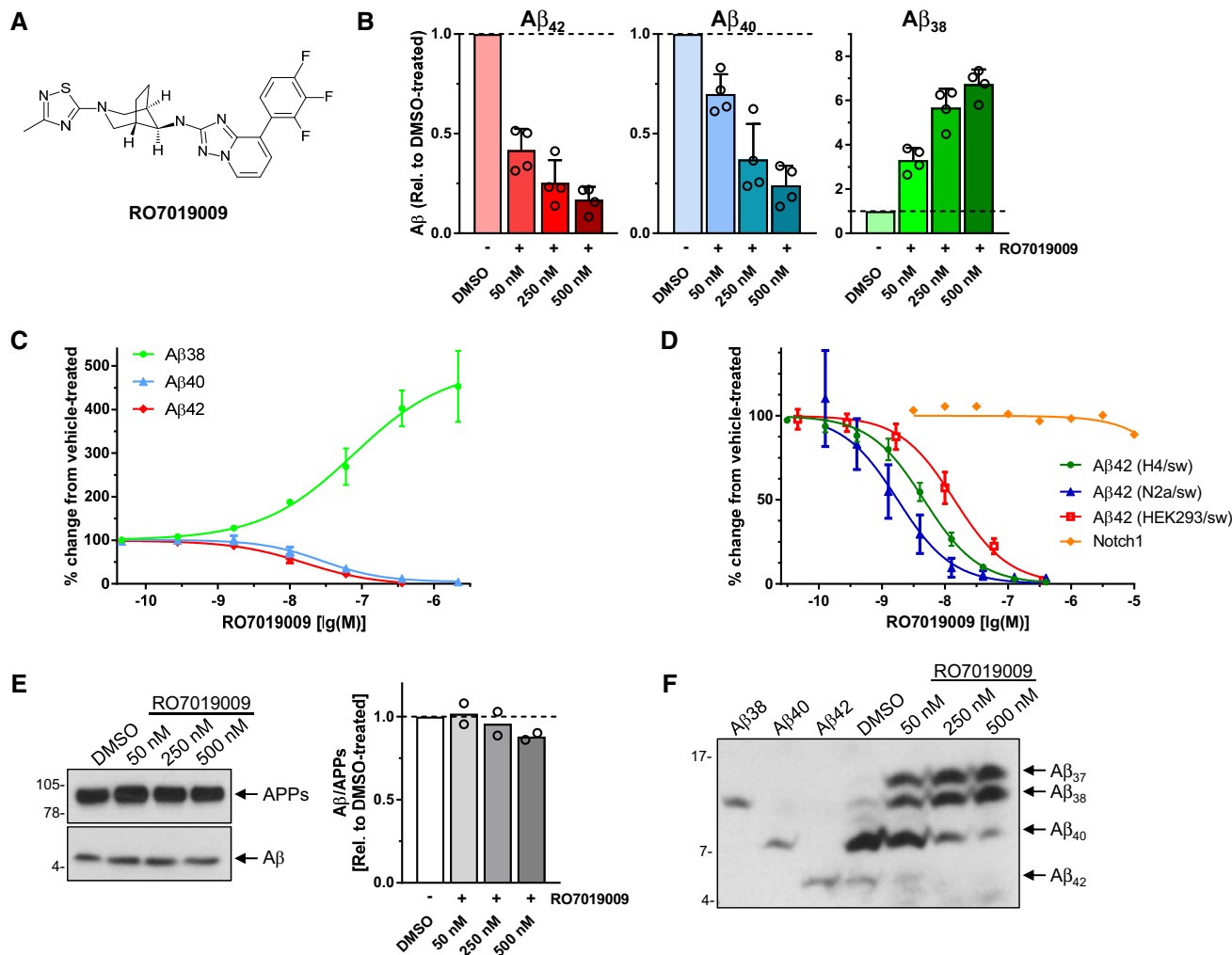

**Figure 3. Characterization of RO7019009.**

A Structure of RO7019009.

B ELISA of Aβ species in conditioned medium of HEK293/sw cells treated with RO7019009 or vehicle (DMSO). Aβ levels are shown relative to DMSO vehicle control (*n* = 4 biological replicates).

C Dose–response curve of RO7019009 in HEK293/sw cells measured using the MSD sandwich immunoassay (*n* = 3 biological replicates).

D Dose–response curves for inhibition of Aβ42 generation and Notch cleavage by RO7019009 in different cell lines. Comparison of *in vitro* potencies for Aβ42 inhibition in HEK293/sw cells, IC$_{50}$ = 14 nM (*n* = 3) (as in (C)) with those in H4/sw cells, IC$_{50}$ = 5 nM (*n* = 7) and N2a/sw cells, IC$_{50}$ = 2 nM (*n* = 6) together with that of Notch cleavage inhibition in a HEK293-based Notch1 luciferase reporter cell line, IC$_{50}$ > 10 μM (*n* = 2) (all *n* numbers represent biological replicates).

E Left panel: Immunoblot analysis of total Aβ in conditioned media of HEK293/sw cells treated with RO7019009 or vehicle (DMSO). Total APPs levels were analyzed to control for normal APP secretion and equal sample loading. Right panel: Quantification of relative Aβ amounts in (E) (*n* = 2 biological replicates).

F Immunoblot analysis of Aβ species in conditioned media of RO7019009-treated HEK293/sw cells after separation by Tris–Bicine urea SDS–PAGE; synthetic Aβ peptides were loaded as size markers.

Data information: Data in (B–F) are presented as mean ± SD.
Source data are available online for this figure.

neuroglioma H4 (IC$_{50}$ (Aβ42) = 5 nM) or mouse neuroblastoma N2a (IC$_{50}$ (Aβ42) = 2 nM) cell lines overexpressing swAPP (H4/sw, N2a/sw) (Fig 3D). Importantly, the signaling cleavage of Notch1 was spared by the compound (IC$_{50}$ (Notch1) > 10 μM) as assessed in a HEK293 cell-based luciferase reporter assay (Fig 3D). As expected for a GSM, RO7019009 did not reduce total Aβ levels (Fig 3E). However, the inhibition of both Aβ40 and Aβ42 generation (Fig 3B and C) indicated that the γ-secretase activity profile

was modulated toward an enhanced processivity in both Aβ product lines. This was confirmed by Tris–Bicine urea SDS–PAGE analysis of the Aβ species which revealed an increased generation of both Aβ37 and Aβ38 with the reduction of Aβ40 and Aβ42 (Fig 3F). The rather strong reduction of Aβ40 with the concomitant increase in generation of Aβ37 indicated that RO7019009 might be potent in lowering Aβ43, which is generated in the Aβ40 product line.

We next tested the response of the mutants to RO7019009 over a concentration range from 50 to 500 nM. As judged from increased Aβ38 levels, measurements of Aβ species by ELISA showed that RO7019009 could enhance the processivity of γ-secretase in all mutants (Fig 4A). Furthermore, specific detection and quantitation of Aβ43 by ELISA showed that RO7019009 reduced the generation of Aβ43 for all mutants including the extreme V261F and R278I mutants, which generated the highest amounts of Aβ43 (Fig 4B). Aβ42 levels could also be efficiently lowered by RO7019009 for several mutants. However, some mutants showed the tendency of a strongly attenuated Aβ42-lowering response (Fig 4C). In particular, the PS1 V261F and R278I mutants were partially resistant to RO7019009. Likewise, Aβ42 levels remained largely unaffected for the L166P mutant (Fig 4C), recapitulating the effect of other potent GSMs on this FAD mutant [24,26,27]. Additional experiments at the 500 nM RO7019009 dose followed by ELISA of Aβ42 levels confirmed a small, yet significant,

reproducible reduction of this species for the less responsive V261F, R278I, and L166P mutants (Fig EV3). These data suggest that the processivity of the V261F, R278I, and L166P mutants can be enhanced by RO7019009 more easily in the Aβ40 product line than in the Aβ42 product line for which much higher GSM concentrations are required. The fact that Aβ43 can be directly processed to Aβ38 [31] might explain the increased levels of Aβ38 of these refractory mutants at lower concentrations of RO7019009.

Mass spectrometry analysis confirmed the reduction of Aβ43 and Aβ42 levels upon treatment with 500 nM RO7019009, including the Aβ42 reduction in the V261F, R278I, and L166P mutants, and also showed a nearly complete reduction of Aβ40 for all mutants (Fig 5). As for the parental HEK293/sw cells (Fig 3F), RO7019009-treated PS1 WT overexpressing cells generated high amounts of both Aβ37 and Aβ38 (Fig 5). This GSM response was also observed for the uncleaved PS1 M292D control and the PS1 V261F and L435F mutants, although

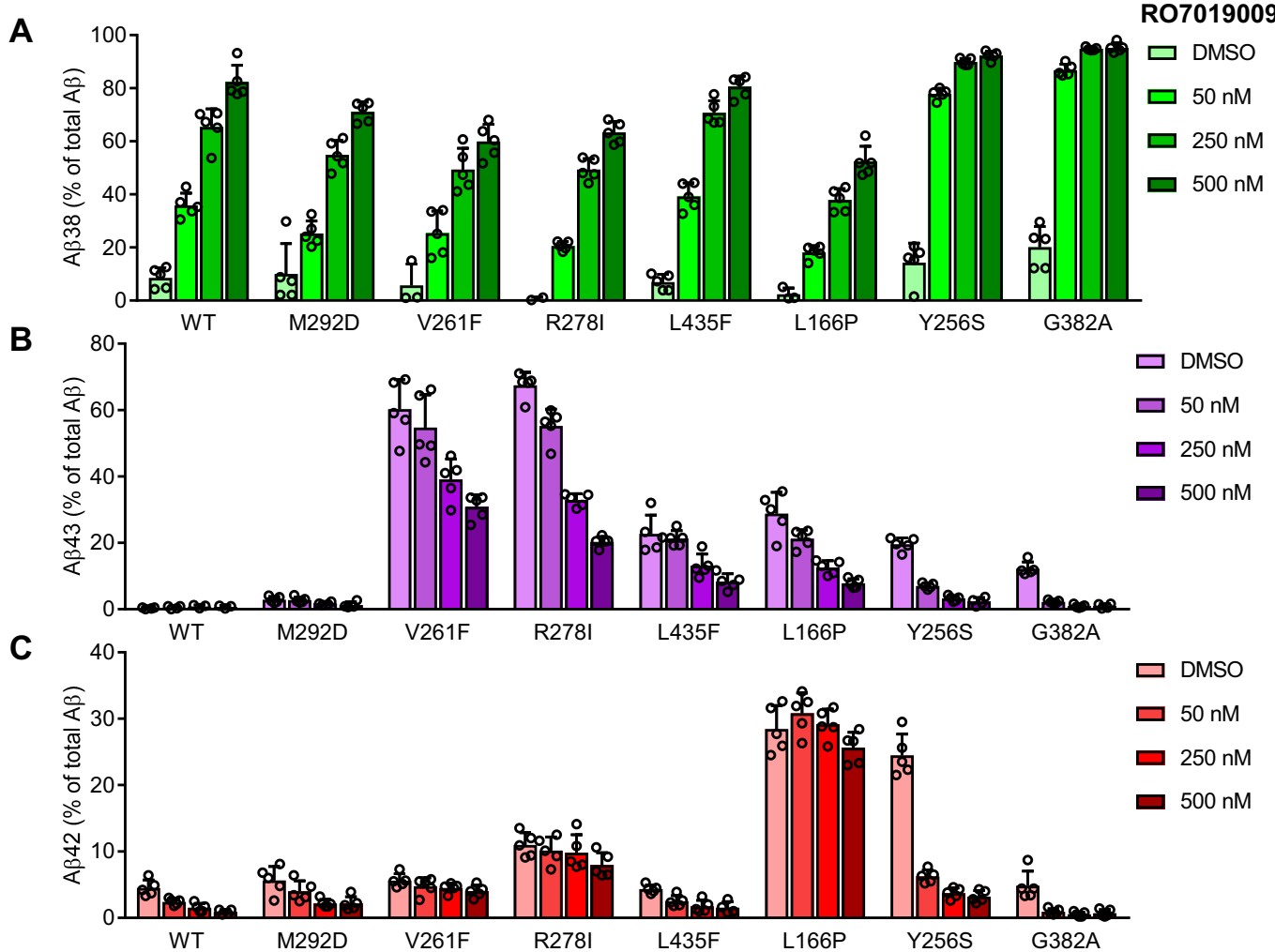

**Figure 4. Modulation of PS1 FAD mutants with RO7019009.**

A–C  ELISA of Aβ species in conditioned media of HEK293/sw cells overexpressing WT or mutant PS1 treated with RO7019009 or vehicle (DMSO) showing the secretion of Aβ38 (A), Aβ43 (B), and Aβ42 (C) compared to total Aβ (Aβ38 + Aβ40 + Aβ42 + Aβ43) (*n* = 5 biological replicates). Note that in some experiments, Aβ38 generation was below the detection limit for some mutants. Data are presented as mean ± SD.

Source data are available online for this figure.

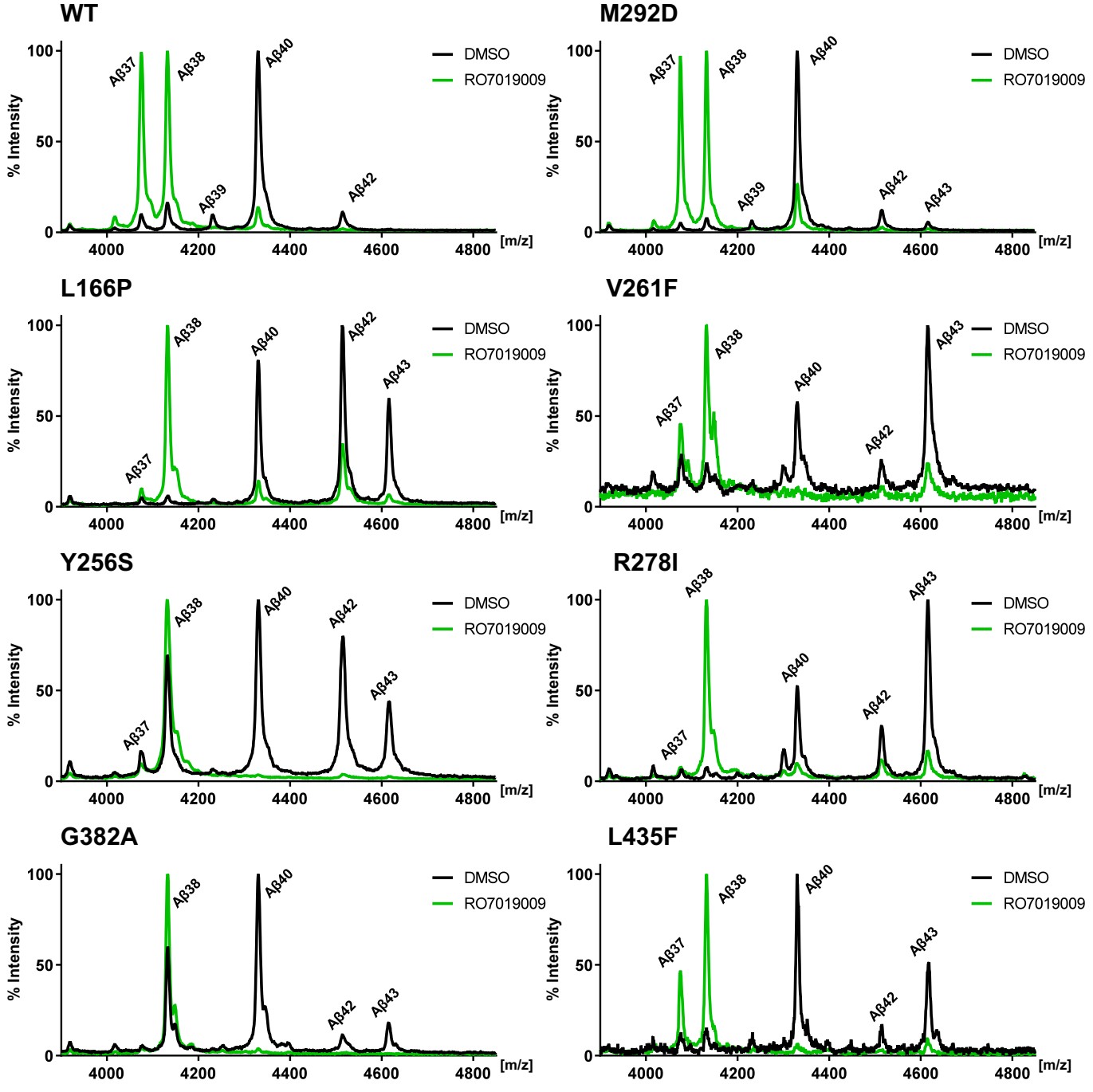

**Figure 5. Modulation of Aβ profiles of PS1 FAD mutants by RO7019009.**

Mass spectrometry analysis of Aβ species immunoprecipitated from conditioned media of HEK293/sw cells overexpressing WT or mutant PS1 treated with 500 nM RO7019009 or vehicle (DMSO).

the two latter mutants generated less Aβ37 than Aβ38. Interestingly, the other mutants generated mostly or nearly exclusively Aβ38 when treated with RO7019009, although longer species of both product lines were reduced. An enhanced direct cleavage of Aβ43 to Aβ38 effected by the GSM thereby skipping the production of Aβ40 and Aβ37 could underlie this observation. Tris–Bicine urea SDS–PAGE analysis provided further independent proof of these findings (Fig EV4).

**Differential effects on substrate crosslinking in the presence of RO7019009**

Changing the interaction of C99 with PS1 could be a potential mode of action of GSMs. To analyze this, C99 V44Bpa was crosslinked to PS1 in the presence of increasing concentrations of RO7019009. As shown in Fig 6A and B, crosslinking of the substrate to WT PS1

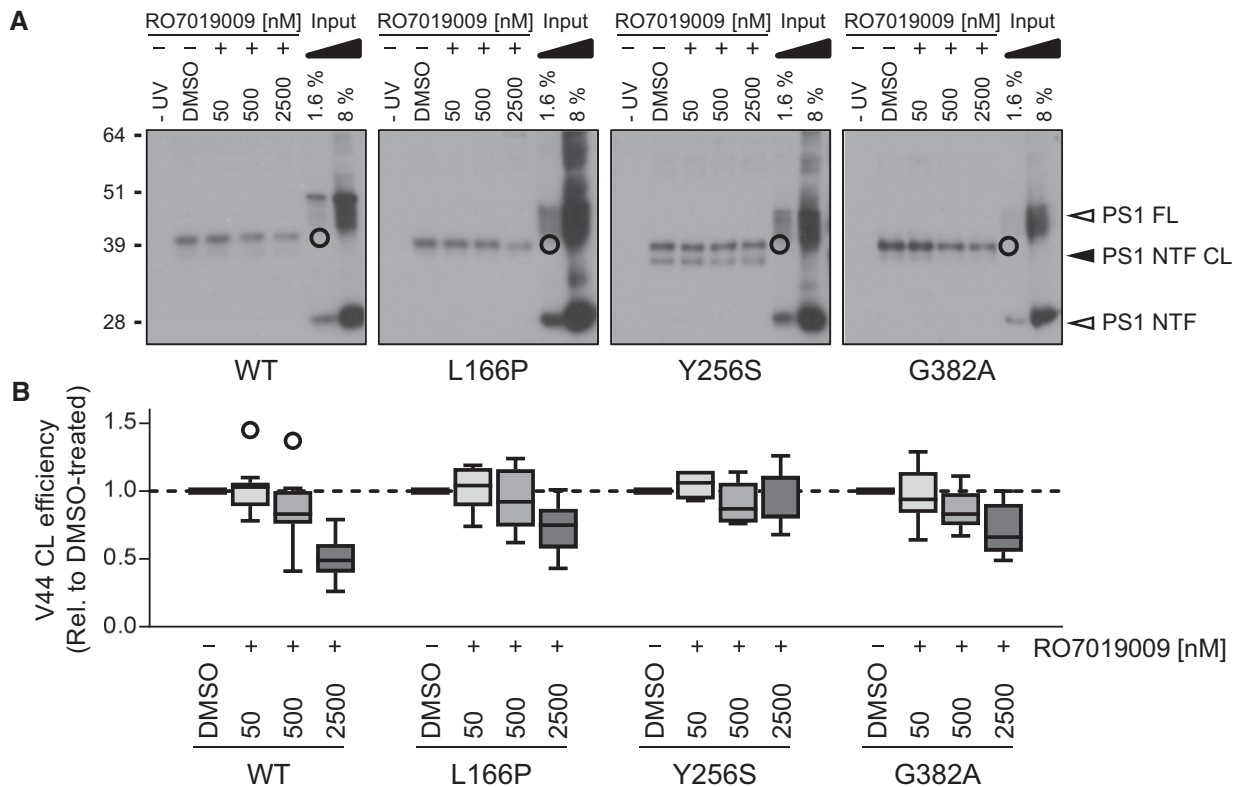

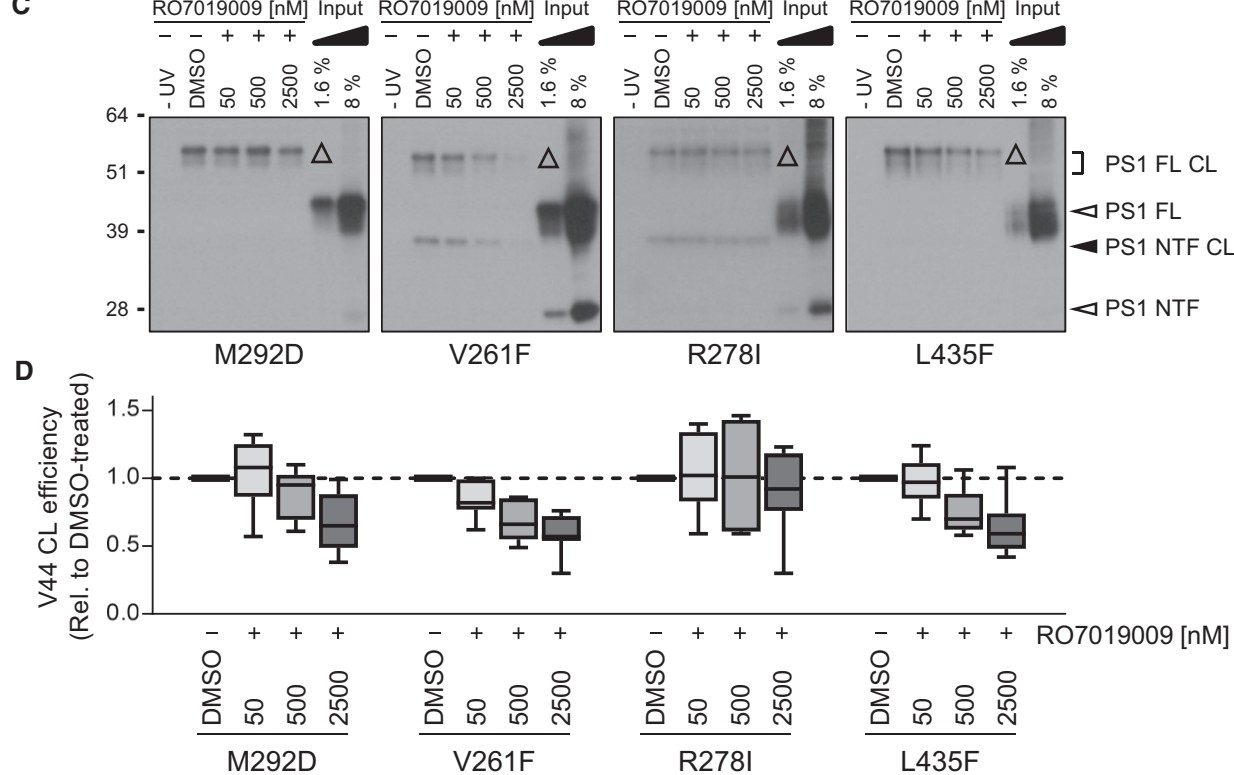

**Figure 6.**

**Figure 6.  Partial effects of RO7019009 on C99–PS1 crosslink formation.**

A   Immunoblot analysis of C99 V44Bpa crosslinking (CL) to PS1 NTFs (circle) of WT or normally endoproteolysed mutants in the presence of vehicle (DMSO) or RO7019009.
B   Quantitation of crosslink efficiencies of C99 V44Bpa to the WT or mutant NTF in (A).
C   Immunoblot analysis of C99 V44Bpa crosslinking to full-length (FL) PS1 (triangle) of endoproteolysis-deficient mutants and the corresponding M292D control in the presence of vehicle (DMSO) or RO7019009.
D   Quantitation of crosslink efficiencies of C99 V44Bpa to the M292 or FAD mutant FL PS1 in (C).

Data information: Data in (B and D) are presented as box plots (line, median; box, 75th–25th percentiles; whiskers, 1.5× interquartile range (IQR); single points represent outliers (> 1.5× IQR away from box)) ($n$ = 7–11 biological replicates).
Source data are available online for this figure.

NTF was reduced in the presence of RO7019009 in a dose-dependent manner. Although this was also the case for several mutants, the effect of the treatment was distinct for every mutant and if clearly visible, only at very high GSM concentrations (Fig 6A–D). Notably, the PS1 Y256S and R278I mutants displayed an unchanged crosslinking. To assess whether these high concentrations of RO7019009 are tolerated by the mutants in the substrate-binding assay, we analyzed their activities to cleave C99 under similar conditions in cell-free cleavage assays. As seen from the increased Aβ37/38 levels, the response to the GSM was maintained by all mutants (Fig EV5A). Moreover, the GSM had little if any inhibiting effect on the initial ε-site cleavages. Even in the presence of 2.5 μM RO7019009, total AICD generation was not or only marginally reduced (Fig EV5B and C). This was not only true for the Y256S and R278I mutants that displayed unchanged crosslinking upon GSM treatment, but also for those mutants that displayed reduced substrate crosslinking. Since reduced overall substrate binding should result in decreased total activity, the unchanged total activity in the presence of the GSM indicates normal overall C99 binding. Thus, the reduced V44 crosslinking levels likely reflect conformational changes of the C99–PS1 substrate–enzyme complex causing altered local substrate interactions. The reduced crosslinking of C99 to the WT may be correlated with the positive response to the GSM in the cell-based assay (Fig 4). However, since C99 crosslinking was not reduced in case of the well responsive Y256S mutant, these data argue against a direct link between reduced C99 crosslinking and GSM-mediated changes in Aβ generation.

## Discussion

Since Aβ43 is increasingly recognized as another toxic species besides Aβ42 in AD pathogenesis, we sought deeper mechanistic insights into the generation of Aβ43 by the PS1 FAD mutants that produce abnormal amounts of it. In addition, we wanted to know whether Aβ43 production by such mutants could be inhibited by GSMs. In profiling and comparing different types of Aβ43-generating PS1 mutants, we observed that mutants that showed a severe loss-of-function phenotype manifesting with strongly reduced total γ-secretase activity and impaired presenilin endoproteolysis produced Aβ43 as prevailing pathogenic Aβ species. This principle was supported by the V261F mutant that was included in our study as a candidate for the generation of Aβ43 based on a previous report that it nearly fully inhibits Aβ generation [32]. On the other hand, the mutants that were cleaved into PS1 NTF and CTF produced lower relative amounts of Aβ43. For these mutants, the levels of Aβ43 could still be higher (PS1 G382A) or lower (PS1 L166P and Y256S)

than that of Aβ42. Of note, the combination of extremely high amounts of Aβ42 with an additional production of Aβ43 in the two latter FAD cases might possibly be causative for or contribute to the very early disease onset in these patients [33,34]. Taken together, our study suggests that the effects on the generation of Aβ43 are rather dramatic when presenilin endoproteolysis is also impaired. A correlation between presenilin endoproteolysis deficiency and increased generation of Aβ43 may have been indicated in an earlier study for several other mutants besides PS1 R278I although the ELISA used at that time could not distinguish Aβ42 from Aβ43 [35]. However, one should note that such a correlation might not always exist as evident from the poorly endoproteolysed PS1 C410Y mutant, which produces ~5-fold more Aβ42 than Aβ43 [18].

Since FAD mutants that generate very high amounts of Aβ43 are often associated with a strong loss of presenilin function, we asked whether such mutants could have even more severe and/or distinctly changed interactions within and around the cleavage domain of C99 compared to mutants that preferentially generate Aβ42. First of all, we found that also the Aβ43-generating mutants clearly changed their interaction with C99, possibly reflecting an altered conformation of the substrate–enzyme complex around the C99 cleavage site region. Thus, altered interactions of C99 with γ-secretase are a general feature of presenilin FAD mutants of various biochemical phenotypes. Mutants that generate abnormal amounts of either Aβ42 and/or Aβ43 interact with C99 differently from WT and independently of whether they undergo presenilin endoproteolysis or not. However, the relative extents of the alterations of C99 crosslinking at the individual positions analyzed were different for the individual mutants. Interestingly, as observed previously for the PS1 A246E and L166P mutants [22], altered crosslinking at T48 and L49, i.e., at the initial substrate cleavage sites of C99, was also observed for the Aβ43-generating mutants suggesting that these could be a common pattern of FAD mutants. In addition, a rather unique crosslinking change of L52 was seen for the G382A mutant and increased crosslinking at K54 even more C-terminal to the ε-sites was found for the poorly endoproteolysed PS1 V261F and R278I mutants, which produced the highest levels of Aβ43. The changes in the substrate interactions caused by the FAD mutants can be rationalized using the recently obtained structure of the APP C83 substrate in complex with the PS1 γ-secretase complex [36]. The FAD mutations and the substrate contact residues analyzed here are all proximal to the active site region (Appendix Fig S2). The positioning alterations of the C99 cleavage region caused by the FAD mutants likely contribute to altered ε-site cleavages that can occur for such mutants [11,37] and could also impact on the ensuing Aβ substrate–γ-secretase interactions. The strengths of these interactions play a key role throughout the carboxy-terminal

trimming pathway and the generation of the pathogenic longer Aβ species [31,38]. In future studies, it would thus be interesting to investigate how FAD mutations would alter interactions of the ensuing Aβ substrates with γ-secretase using Bpa-mediated photoaffinity mapping.

A major question of this study was whether Aβ43 could be lowered by GSMs, especially when generated in abnormal amounts by particular PS1 FAD mutants. Previous *in vitro* studies including patient-derived neuronal cells showed that Aβ42 could be lowered for many presenilin FAD mutants by potent GSMs [26–28] opening treatment possibilities, for example, within the Dominantly Inherited Alzheimer Network (DIAN) [39], based on a rational selection of a GSM effective for a given presenilin FAD mutation. We now show that Aβ43 production can also be inhibited by modulation of γ-secretase activity. We identified RO7019009 as a potent GSM with CNS drug-like properties, which could lower Aβ43 generation in all investigated mutants. These include the PS1 R278I and PS1 L166P mutants for which the well-characterized GSMs RO-02 and GSM-1 showed strongly reduced efficacy as compared to PS1 WT. However, although RO7019009 could efficiently inhibit the generation of Aβ43 in all the mutants, remarkably, for some of the mutants including the strong Aβ43-overproducing PS1 mutants V261F and R278I, their concomitant Aβ42 production could only be inhibited at higher RO7019009 concentrations and only to small extents. The same observation was also made for the L166P mutant, but not for the Y256S mutant, which has a very similar Aβ profile as the L166P mutant. For the PS1 Y256S mutant, production of both Aβ42 and Aβ43 could be efficiently inhibited at low RO7019009 concentrations. In addition, generation of the shorter Aβ species was differentially affected by RO7019009 in the various mutants. Some mutants were modulated in a way that increased levels of both Aβ37 and Aβ38, while others showed only minor or no generation of Aβ37 while still producing high levels of Aβ38. These observations suggest that RO7019009 differentially affects the two product lines in certain mutants resulting in, e.g., less effective Aβ42 reduction or generation of predominantly Aβ38.

GSMs have been shown to reduce the dissociation of Aβ42–γ-secretase complexes and increase their stability [31,38]. The resulting longer substrate residence time would thereby allow more efficient carboxy-terminal processing toward shorter Aβ species. Mutational analysis further showed that the activity of GSMs is affected by K28 and nearby residues of the extracellular TMD border of C99 [40–42]. As shown very recently, these effects relate functionally to the proximity of K28 to NCT [36,43] and indicate this contact region with C99 and/or Aβ also as part of a GSM binding site [44].

Since it remained possible that RO7019009 may exert its activity by affecting the interaction of C99 with γ-secretase, we probed the crosslinking of V44, which represents the position of C99 that shows the most efficient crosslink in the PS1 NTF [22]. While two mutants did not change crosslinking in the presence of the GSM, it was decreased for WT PS1 and most mutants, although to different extents. Notably, total γ-secretase activity was unaffected by the GSM. Thus, the crosslinking changes induced by RO7019009 seem to be due to a slightly changed substrate–enzyme complex conformation causing altered local substrate docking rather than decreased overall substrate binding. However, since clear effects of allosteric modulation by RO7019009 at this major interaction site of

γ-secretase were observed only at very high concentrations of the GSM, it is probable that these effects are not relevant for the activity of the GSM. Rather, the interactions with γ-secretase could be changed for long Aβ species, which become the ensuing substrates after C99 has been cleaved [31,38]. Effects on the interactions of further processed Aβ species originating from Aβ48/49 with the protease seem even more likely since GSMs do not change cleavage of C99 at the ε-sites [29,45]. Conformational changes in the presenilin active site region upon GSM binding that have been reported earlier [46] may therefore affect γ-secretase at this level. Thus, the altered crosslinking efficiencies of C99 observed here could more likely reflect a broader conformational enzyme change induced by the GSM, supporting the idea that GSMs rather act on the sequential carboxy-terminal cleavage of the longer Aβ42/43 species.

In conclusion, our study shows that all investigated Aβ43-generating PS1 FAD mutants affect the interaction of the C99 cleavage site region with γ-secretase. Altered C99–γ-secretase interactions are thus a general feature of presenilin FAD mutants reinforcing the concept that such changes are critically contributing to the mechanism of the pathogenic activity of FAD caused by presenilin mutations. As an important implication for AD therapy approaches, we conclude that Aβ43-generating FAD mutations should in principle be targetable by γ-secretase modulation, which could be beneficial for such mutation carriers.

# Materials and Methods

### Antibodies

Antibodies to the PS1 CTF (3027 [47], immunoblot (IB): 1:4,000), the PS2 CTF (BI.HF5c [23], IB: 1:2,000) as well as antibodies to total Aβ (3552 [48], immunoprecipitation (IP) 1:500 and 2D8 [49], IB 3 µg/ml) have been described previously. Monoclonal antibody 2G7 (IgG2b/k) (IB 3 µg/ml) to the PS1 NTF was raised in C57/BL6 mice against residues 39–52 (NDRRSLGHPEPLSN) of human PS1. Antibodies N1660 to NCT (Sigma N1660, IB 1:1,000), 22C11 to secreted soluble APP (Merck Millipore MAB348, IB 1:5,000), 4G8 to Aβ (Covance SIG-39220, IP 1:500–1:2,500), Y188 to the APP C-terminus (Abcam ab32136, IB 1:4,000), and Penta-His (Qiagen 34460, IB 1:2,000) were obtained from the indicated companies. Species-specific anti-Aβ antibodies to Aβ40 (BAP24) and Aβ42 (BAP15) were kindly provided by Manfred Brockhaus (Roche Applied Science) and SULFO-tagged according to the instructions of the supplier (Meso Scale Discovery (MSD)). The anti-Aβ37 antibody (D2A6H) was obtained from Cell Signaling (#12467S) and SULFO-tagged as above. The SULFO-tagged antibody against Aβ38 was obtained from MSD.

### cDNA constructs

PS1 mutations were introduced into pcDNA3.1/Zeo(+) expression vector containing the WT PS1 cDNA by site-directed mutagenesis and verified by sequencing.

### Cell lines and cell culture

Single-cell clones of HEK293 cells stably co-expressing swAPP with WT and mutant PS1 constructs were cultured as described [50]. For

GSM treatments, the cells were grown to a confluence of 70–80%. Thereafter, medium was changed to fresh medium containing either DMSO vehicle control or GSM at the indicated concentration and the cells were incubated for 16–18 h before Aβ analysis of conditioned medium. Single-cell clones of H4 overexpressing swAPP and HEK293-based Notch1 reporter cell lines were cultured and treated with GSM as described [51]. Mouse N2a cells stably overexpressing swAPP were cultured in DMEM (high glucose) supplemented with 10% FCS and 1 mM sodium pyruvate and treated with GSM as described above.

### GSMs

GSM-1 and RO-02 have been described previously [24,29]. RO7019009 was synthesized as depicted in Fig EV2.

### Protein analysis

Immunoblot analysis of PS1 and PS2, NCT, full-length APP, C-terminal APP fragments, secreted APPs, and Aβ was performed as described [52]. To analyze GSM dose responses and determine $IC_{50}$ values for Aβ42 inhibition, secreted Aβ species of HEK293/sw cells were analyzed by sandwich immunoassay (MSD) using SULFO-tagged C-terminal specific antibodies to Aβ38, Aβ40, and Aβ42 as described before [26] or, in case of H4/sw and N2a/sw cells by an Aβ42 AlphaLISA immunoassay kit according to the instructions of the supplier (PerkinElmer). Notch1 cleavage reporter assay was carried out as described [51]. GSM-mediated changes of secreted Aβ species in conditioned medium of PS1 WT or mutant expressing HEK293/sw cells were quantified by ELISA using end-specific C-terminal antibodies to Aβ38, Aβ40, Aβ42, and Aβ43 obtained from IBL as described previously [16]. Mass spectrometry analysis of secreted Aβ species was performed after immunoprecipitation using antibody 4G8 as described previously [17]. Individual Aβ species were analyzed by immunoblotting, using Tris–Bicine urea SDS–PAGE [53]. To improve the separation of longer Aβ species (e.g., Aβ42/43), the separation gel was adjusted to 8% acrylamide and 8 M urea. Aβ37 and Aβ38 generated in cell-free γ-secretase assays were analyzed by sandwich immunoassay (MSD) as above using SULFO-tagged species-specific antibodies. AICD generation in these assays was analyzed by immunoblotting with Penta-His antibody after sample separation on 10–20% Tris–Tricine gels (Invitrogen). Protein bands from immunoblots were quantified using the LAS-4000 image reader (Fujifilm Life Science) and Multi-Gauge V3.0 software for analysis.

### Substrate photocrosslinking

Bpa-containing C99-based γ-secretase substrates were prepared as described previously [22]. CHAPSO-lysates were prepared from HEK293/sw cells co-expressing either PS1 WT or PS1 mutants. Photocrosslinking was performed as described before [22]. Crosslinking of the substrates was analyzed by immunoblotting using antibodies against the PS1 NTF as described [22]. For substrate crosslinking experiments in the presence of RO7019009, the cell lysates were pre-incubated for 30 min on ice with RO7019009 or DMSO vehicle before the addition of C99 V44Bpa and subsequent photocrosslinking.

### Cell-free γ-secretase assay

Membrane fractions were prepared from HEK293/sw cells co-expressing WT or mutant PS1 as described before [54]. γ-secretase was solubilized from the membrane fractions with 1% CHAPSO and used for cell-free cleavage assays with recombinant C100-His$_6$ as substrate as described [17].

**Expanded View** for this article is available online.

### Acknowledgements

This work was supported by the DFG (FOR2290) (HS). We thank Alice Sülzen for technical assistance with antibody generation and characterization.

### Author contributions

JT and HS conceived and designed experiments. JT performed experiments. RMRS and KB synthesized and characterized RO7019009. KB provided additional GSMs. AF provided materials and guidance for substrate crosslinking experiments. RF generated monoclonal antibody 2G7. JT, AF, and HS analyzed data and interpreted results. HS supervised the project and wrote the paper with contributions from JT, RMRS, and KB.

### Conflict of interest

RMRS and KB are employees of F. Hoffmann-La Roche Ltd. The other authors declare no conflict of interest.

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
