## [Review Process File · EMBO Reports]

A β 43-producing PS1 FAD mutants cause altered substrate interactions and respond to γ -secretase modulation

Johannes Trambauer, Rosa María Rodríguez Sarmiento, Akio Fukumori, Regina Feederle, Karlheinz Baumann, Harald Steiner

Review timeline:

Submission date:	25 February 2019
Editorial Decision:	16 May 2019
Revision received:	20 September 2019
Editorial Decision:	21 October 2019
Revision received:	28 October 2019
Accepted:	29 October 2019

Editor: Esther Schnapp

Transaction Report:

1st Editorial Decision

16 May 2019

Thank you for your patience while your manuscript was peer-reviewed at EMBO reports. I am very sorry for the unusual delay in getting back to you. We were still waiting for the third referee report to come in, but despite several reminders, we did not hear back from referee 1, so I decided to make a decision now based on the 2 reports we have, and on referee 3's cross-comments, all pasted below.

As you will see, referee 2 raises critical points, but referee 3 agrees that you should be given a chance to respond to these. I would therefore like to invite you to revise your manuscript with the understanding that the referee concerns must be fully addressed and their suggestions taken on board. Please address all referee concerns in a complete point-by-point response. Acceptance of the manuscript will depend on a positive outcome of a second round of review. It is EMBO reports policy to allow a single round of revision only and acceptance or rejection of the manuscript will therefore depend on the completeness of your responses included in the next, final version of the manuscript.

I look forward to seeing a revised version of your manuscript when it is ready. Please let me know if you have questions or comments regarding the revision.

REFeree REPORTS

Referee #2:

In this study, Trambauer and colleagues examine whether the A β 43-increasing R278I, L435F and V261F PS1 mutants alter the binding of APPC99 to the gamma secretase complex by applying an established photocrosslinking approach.

This approach relies on the incorporation of the photoactivable unnatural amino acid (p-benzoylphenylalanine, Bpa) at selected positions of the APPC99 substrate. The positions were

determined previously, by the same group, as the residues establishing most prominent crosslink with the substrate-binding site of PS1.

Here, the authors generated HEK293 cell lines stably overexpressing Swedish (sw) mutant APP (HEK293/sw) together with the A β 43-increasing and endoproteolysis deficient R278I, L435F and V261F PS1 mutants. The uncleavable M292D PS1 (active as full length PS1) mutant was used as a control. In addition, the normally endoproteolysed L166P and Y256S FAD mutants, which generate high amounts of A β 42 and A β 43, as well as the artificial G382A FAD-like mutant, generating more A β 43 than A β 42 were analyzed.

The crosslinking experiments revealed that the FAD PS1 mutants differ in their pattern of interaction with the substrate, when compared to the WT or the M292D PS1 control. However, the crosslinking alterations exhibited by these mutants did not clearly correlate with the associated tendencies to generate A β 43 or A β 42.

Given that A β 43 is the most abundant peptide generated by the R278I, L435F and V261F PS1 variants, the authors also investigated whether compounds referred to as gamma secretase modulators (GSMs) lower its production.

The authors showed that A β 43 production by the FAD R278I, L435F and V261F mutants was lowered by the potent RO7019009 GSM. However, the presented data suggest that this compound fails to reduce A β 42 generation by the R278I and V261F mutants. Similarly, the GSM decreased A β 43, but did not change A β 42 production by the L166P variant. In contrast, it lowered the generation of both A β 43 and A β 42 by the Y256S and G382A mutants. In the final part of the study, the authors investigated the effect of the RO7019009 GSM on the interaction between APPC99 and PS1-NTF. Crosslinking analysis of APPC99 V44Bpa to PS1-NTF revealed that "binding" of the substrate to the WT PS1-NTF was reduced by the GSM in a dose dependent manner. Intriguingly, the GSM had differential effects on the FAD-linked mutants: APPC99/PS1-NTF crosslinking was reduced to a distinct degree for all but Y256S and R278I cases.

Overall, the data do not reveal clear correlations between substrate binding and the generated A β profiles. The authors thus discuss distinct scenarios.

Major comments:

-The crosslinking data presented in this study support the conclusion that FAD-linked mutations alter the interaction between the APPC99 substrate and the gamma secretase complex. The findings are intriguing but remain descriptive. The most important point is the lack of direct biochemical analysis (i.e. affinity constants) demonstrating that the differential crosslinking patterns actually reflect altered substrate binding as the authors conclude.

-The authors discuss that the altered patterns could be linked to the changes in the strengths of the interactions between PS1 and APPC99. Importantly, the reduction in the strength of the gamma secretase-APPC99 binding has been shown in two independent studies to play a key role in the promotion of the generation of longer A β species (Okochi et al., 2013, Szaruga et al., 2017). However, the referred studies indicate that pathogenic PS1 mutations affect substrate dissociation rather than its binding affinity. What is the view of the authors on this point? How can they put together the different findings?

-The crosslinking of APPC99 V44Bpa to PS1-NTF is reduced by the GSM RO7019009 in a dose dependent manner. Does these results actually support decreased substrate binding or changes in the mode of docking of the substrate relative to PS1-NTF? (the distance between the V44Bpa residue and the PS1-NTF fragment does not facilitate crosslinking)

-The authors propose that the RO7019009 GSM may specifically affect the A β 40 product line of the gamma secretase. However, this conclusion is not fully supported by the data. Kinetic investigation of the effects of this GSM on the gamma secretase proteolysis is needed in order to solve the intriguing findings.

Referee #3:

This report further evaluates how Presenilin mutations that are linked to FAD alter gamma-secretase function and respond to small-molecules that shift gamma-secretase cleavage (GSMs). The main novelty of the data are the findings that a new GSM can still shift cleavage of severe PS1 mutants that primarily make A β 43, that PS1 mutants which generate mostly A β 43 are less functional in overall cleavage than other FAD mutants that make more A β 42, that these mutants and the GSM shift the contact sites of C99 with gamma-secretase. The data is clearly presented and supports the conclusions. There are few issues with the discussion and some other minor issues.

- 1) There are several "models" that have been put forward regarding how GSMs work and how gamma-secretase cleaves in addition to those cited. It would be useful to include some discussion of these and how the current work.
- 2) I see there is an author from Roche, but little other information is provided on the new GSM. Is it a clinical candidate? It would be nice to show that it is in fact brain penetrant and can lower Abeta in mice or some other preclinical model.
- 3) I have never liked the loss of function description for Presenelins but favor the term shift in function. As these authors know there is really no evidence that FAD-linked PS1 mutants are loss of function in humans. This is more of a personal preference, but the whole loss of function PS hypothesis really did not really advance science in this area but was somewhat of a distraction....

Cross-comments from referee 3:

I see the other comments are less enthusiastic.

I think that my own thought is that the following is reasonable to ask for the authors to address, but I still think they deserve a chance to respond.

-The authors propose that the RO7019009 GSM may specifically affect the A β 40 product line of the gamma secretase. However, this conclusion is not fully supported by the data. Kinetic investigation of the effects of this GSM on the gamma secretase proteolysis is needed in order to solve the intriguing findings.

-The authors discuss that the altered patterns could be linked to the changes in the strengths of the interactions between PS1 and APPC99. Importantly, the reduction in the strength of the gamma secretase-APPC99 binding has been shown in two independent studies to play a key role in the promotion of the generation of longer A β species (Okochi et al., 2013, Szaruga et al., 2017). However, the referred studies indicate that pathogenic PS1 mutations affect substrate dissociation rather than its binding affinity. What is the view of the authors on this point? How can they put together the different findings?

I think this request is a tough one, one can get enzyme kinetics but I am skeptical about affinity measurements between CTF and gamma-secretase. I am not aware of anyone doing this affinity measurement, and if they have I am not sure that it would be meaningful. Indeed, I don't really know how one does such a measurement in a detergent containing solution. I think that is why the authors used this cross-linking approach.

-The crosslinking data presented in this study support the conclusion that FAD-linked mutations alter the interaction between the APPC99 substrate and the gamma secretase complex. The findings are intriguing but remain descriptive. The most important point is the lack of direct biochemical analysis (i.e. affinity constants) demonstrating that the differential crosslinking patterns actually reflect altered substrate binding as the authors conclude.

1st Revision - authors' response

20 September 2019

(Please see next page)

Thank you for handling our manuscript “**A β 43-producing PS1 FAD Mutants cause substrate-binding changes and respond to γ -secretase modulation**”, which we would like to resubmit to EMBO Reports as a revised version.

We first want to thank both referees for their fair and constructive reviews of our manuscript, which we felt were overall quite encouraging. While referee 3 gave a very positive evaluation of our findings, found our conclusions well supported by the data and raised only a few points mostly regarding the discussion, reviewer 2 appreciated our principal observations but was somewhat skeptical regarding our conclusions and had a few concerns.

In our revised version, we have now addressed all the points raised by the two referees considering also the cross-comments of referee 3. As you will see in our response below, all comments were fully taken into consideration and addressed. As a result, additional data were provided (**Figures EV3, EV4, and EV5**) and several text passages of the manuscript were rewritten in our revised version. Please note that these changes were highlighted in yellow in an additional file accompanying our revised manuscript. We also exchanged the A β mass spectrum of the PS1 L435F mutant for a better quality one in **Figure 1C**, increased the number of data points in **Figure 4** (now $n = 5$) and provide a structural presentation of presenilin/ γ -secretase with an APP substrate fragment (Zhou et al., 2019) depicting the FAD mutations and the crosslink residues investigated and showing their relatively close spatial vicinity to the active site (optional **Appendix Figure S2**).

Referee #2: Major comments:

-The crosslinking data presented in this study support the conclusion that FAD-linked mutations alter the interaction between the APPC99 substrate and the gamma secretase complex. The findings are intriguing but remain descriptive. The most important point is the lack of direct biochemical analysis (i.e. affinity constants) demonstrating that the differential crosslinking patterns actually reflect altered substrate binding as the authors conclude.

Response:

The referee criticizes that we concluded that the differentially altered C99 substrate interactions with γ -secretase by the FAD mutants reflected altered substrate binding without providing evidence for this by direct measurements of binding affinity constants. After carefully inspecting all relevant text passages, we assume that there is a misunderstanding of what we intended to say with the term “altered substrate binding”. Our data clearly show that the interactions are altered at several prominent substrate positions and we have used the term “altered substrate binding” strictly only in this sense. They were not meant to imply that also overall binding affinities are changed, and as they also do not necessarily imply this, we did not conclude this from our data.

In that regard, it is important to note that there is no evidence in the literature for altered CTF substrate affinities of presenilin FAD mutants. Assays that could directly measure affinity constants between C99 and γ -secretase are not established in the field and may also be hampered by inherent methodological problems, as noted also by referee 3 in his cross-comment. There is however one study in which K_d values were derived for a somewhat different scenario – initial binding of C99 from an indirect substrate-binding competition assay in the presence of an active-site directed γ -secretase inhibitor, but no differences were noted between wt PS1 and FAD mutants (Szaruga et al., 2017). Previous measurements of K_m values did also not indicate major differences in substrate binding of PS1 FAD mutants towards C99 and other γ -secretase substrates (Chavez-Gutierrez et al., 2012). Thus, there is no indication that presenilin FAD mutants would change substrate binding affinities.

Besides the problem that direct measurements of affinity constants are not feasible, which was also noted in the cross-comment of referee 3, we would respectfully also argue that such experiments are beyond the scope of this study and will in fact also not change our principal observation that the substrate interactions are changed by the FAD mutants for prominent crosslink sites of C99, which was

also appreciated by referee 2. However, to avoid a misunderstanding of what was meant with “altered substrate binding” we reworded the text in the revised manuscript at several places.

-The authors discuss that the altered patterns could be linked to the changes in the strengths of the interactions between PS1 and APPC99. Importantly, the reduction in the strength of the gamma secretase-APPC99 binding has been shown in two independent studies to play a key role in the promotion of the generation of longer A β species (Okochi et al., 2013, Szaruga et al., 2017). However, the referred studies indicate that pathogenic PS1 mutations affect substrate dissociation rather than its binding affinity. What is the view of the authors on this point? How can they put together the different findings?

Response:

We thank the referee for bringing up the question of how differences in the interactions of C99 with PS1 FAD mutants shown in our study could influence the generation of longer pathogenic A β species, which we, as we realize now, may not have been optimally described in the discussion of the original manuscript.

Before explaining our view on this, it is important to note that the referee may have missed a critical point: we did not mean that the altered crosslink patterns may be linked to altered affinities between C99 and γ -secretase, but rather that they could also impact on the subsequent interactions of the A β species with the enzyme. In addition, the Okochi et al. and Szaruga et al. studies reported reduced binding strengths/affinities between A β substrates but as the Szaruga et al. study shows, not for C99 as the referee stated. Both studies in fact suggest that substrate residence time at the γ -secretase complex, after C99 has been cleaved at the ϵ -sites, is the important factor for the subsequent carboxy-terminal trimming of A β 49/48 down to the shorter A β species (i.e. the processivity of γ -secretase). The longer the A β substrate intermediates can stay bound, the more efficient they will be processed down to short A β species. In case of FAD mutants in presenilin, longer species such as A β 42/43 species will fall off from the enzyme prematurely resulting in pathogenic A β ratios.

We thus envision that while affinity changes of A β species (i.e. enhanced off rates due to impaired A β - γ -secretase substrate enzyme complex stabilization) will mainly affect the processivity (Okochi et al., 2013, Szaruga et al., 2017), the binding changes described in our study, which are in fact positioning changes of the C99 cleavage domain, will likely first induce changes on ϵ -site cleavage selection, which is known to dictate A β product line selection (A β 49/46/43/40 and A β 48/45/42/38). However, since the substrate is already mispositioned for the first cleavage, these alterations could impact also on the ensuing cleavages. Since it may have not been clear enough described in the original manuscript, we have rewritten the respective part in the discussion.

-The crosslinking of APPC99 V44Bpa to PS1-NTF is reduced by the GSM RO7019009 in a dose dependent manner. Does these results actually support decreased substrate binding or changes in the mode of docking of the substrate relative to PS1-NTF? (the distance between the V44Bpa residue and the PS1-NTF fragment does not facilitate crosslinking)

Response:

In principal, both scenarios – decreased overall substrate binding or altered mode of docking are possible. Since for some mutants the interaction was not changed, i.e. the crosslink efficiencies of C99 V44Bpa were not reduced by the GSM, we would favor the idea that the interactions can be affected by conformational changes induced by the GSM resulting in some cases (e.g. for wt and some of the mutants) in altered docking and thus in reduced crosslinking. Generally, altered crosslinking efficiency (increased or decreased V44Bpa crosslink efficiency) does not necessarily reflect a global affinity change but certainly a locally altered interaction at this particular site. In line with these considerations, it was also reported in the Szaruga et al., 2017 study that the binding affinity of C99 was not changed for PS1 FAD mutants in the presence of a GSM in their substrate-binding competition assay.

In our revised version, we have added an additional supplementary figure, **Figure EV5**, showing that modulator activity (**Figure EV5A**) and total γ -secretase activity (**Figure EV5B and C**) is maintained even at very high compound concentrations in an *in vitro* cleavage assay under conditions closely resembling that used for substrate crosslinking. Reasoning that normal activity should only be observed

when overall substrate binding is unaffected, these data would additionally argue that overall C99 binding is quantitatively normal even at very high concentration of the GSM. Thus, the GSM could instead change the conformations of wt PS1 and some of the mutants and/or the enzyme-substrate complexes in a way such that the crosslink efficiency can change for C99 V44Bpa.

-The authors propose that the RO7019009 GSM may specifically affect the A β 40 product line of the gamma secretase. However, this conclusion is not fully supported by the data. Kinetic investigation of the effects of this GSM on the gamma secretase proteolysis is needed in order to solve the intriguing findings.

Response:

We agree that the differential actions of the GSM on the A β product lines are indeed intriguing. Our statement that RO7019009 may act as a A β 40-product-line specific GSM related, however, only to certain mutants such as PS1 R278I and L166P (for which A β 43 but not A β 42 could be lowered), as we had written in the original manuscript. Clearly, as was also stated in the original manuscript, the proposed “product line specificity” is not generally observed as both A β 40 and A β 42 product lines are modulated for other mutants, as most clearly seen for example for the PS1 Y256S mutant.

Due to the reviewer’s concern whether there is really such a “product line specificity”, we reanalyzed our data and performed additional experiments to investigate the modulatory effects of RO7019009 in more detail. Since assays which could test the kinetics of conversion of different A β species within specific product lines could not be established in our laboratory, we resorted to additional A β ELISA as well as mass spectrometry analyses, which reveal the full profile of A β species. These experiments showed that A β 42-generation in the PS1 R278I and L166P mutants can be reduced at higher concentrations (500 nM) of the GSM, i.e. the GSM acts also on the A β 42-product line, although the effect is still less pronounced when compared to the reduction of A β 43 levels coming from the A β 40-product line. This trend was already visible in the data of the original manuscript (500 nM, Figure 4C) but unfortunately not recognized. Therefore, we conclude that the proposal of a product line specific action of RO7019009 in the original manuscript was somewhat too strongly stated and as the reviewer correctly recognized not fully supported by our data. We thank the reviewer for drawing our attention to this.

Interestingly, the additional mass spectrometry experiments showed differential effects of RO7019009 on the production of A β 37 and A β 38 in the different mutants. These distinct changes elicited by RO7019009 could be explained by mutant-specific changes in product line usage and/or crossing.

Since our additional analyses clarify the issue raised by the reviewer and provide interesting more detailed information on the A β profiles in the different mutants in the presence of RO7019009, we have added these data to our revised manuscript as **Figures EV3, EV4A and EV4B** and amended the text of the manuscript at the relevant places accordingly.

Referee #3:

This report further evaluates how Presenilin mutations that are linked to FAD alter gamma-secretase function and respond to small-molecules that shift gamma-secretase cleavage (GSMs). The main novelty of the data are the findings that a new GSM can still shift cleavage of severe PS1 mutants that primarily make Abeta43, that PS1 mutants which generate mostly abeta43 are less functional in overall cleavage than other FAD mutants that make more Abeta42, that these mutants and the GSM shift the contact sites of C99 with gamma-secretase. The data is clearly presented and supports the conclusions. There are few issues with the discussion and some other minor issues.

Response:

We thank the referee for highlighting and appreciating the main novelty of our data. He/she felt that the data support our conclusions and had only a few remaining issues.

1) There are several "models" that have been put forward regarding how GSMs work and how gamma-secretase cleaves in addition to those cited. It would be useful to include some discussion of these and how the current work.

Response:

We thank the reviewer for the suggestion to also elaborate somewhat on GSM mechanisms, which indeed was not much touched in the original manuscript. We have thus included a few sentences in the discussion describing how GSMs might work covering the currently proposed principal mechanisms that suggest reduced dissociation and/or stabilization of A β 42/43- γ -secretase complexes (Okochi et al., 2013, Szaruga et al., 2017). These studies basically indicate an increased residence time of long A β on the enzyme such that it can be processed more efficiently to the shorter species. Also, residues of the luminal transmembrane domain border of C99 such as K28, which is thought to facilitate substrate-anchoring by its interaction with nicastrin (Petit et al. 2019), were reported to be critical for the mechanism and pharmacology of GSMs (Page et al., 2008, Ousson et al., 2013, Jung et al., 2014). These studies, which additionally suggest modulation of substrate-anchoring as a potential mode of action of GSMs, were also described in the revised discussion.

2) I see there is an author from Roche, but little other information is provided on the new GSM. Is it a clinical candidate? It would be nice to show that it is in fact brain penetrant and can lower Abeta in mice or some other preclinical model.

Response:

As suggested by the reviewer, we have added a few more additional data regarding the properties of RO7019009. Since it was not the focus of our present study, we hope that the referee will find it acceptable that we will not comment on the status or on the *in vivo* properties of RO7019009 in our revised version. These data shall be presented in another study focusing on the discovery and *in vivo* profiling of GSMs with RO7019009 showing a potent A β -lowering behavior in a mouse model. However, what we can already additionally report here in our revised manuscript is that RO7019009, besides its high *in vitro* potency, has favorable CNS drug-like properties that predict good brain penetration such as acceptable lipophilicity (logD: 4.00), solubility (3 μ g/ml) and high permeability (PAMPA: high) combined with very low metabolic clearance (rate of elimination from the body) in human and mouse (Clmicr human/mouse, μ l/min/mg: <10/<10 (low/low)). These data are now presented in the revised version as **Table EV1**.

3) I have never liked the loss of function description for Presenelins but favor the term shift in function. As these authors know there is really no evidence that FAD-linked PS1 mutants are loss of function in humans. This is more of a personal preference, but the whole loss of function PS hypothesis really did not really advance science in this area but was somewhat of a distraction....

Response:

We fully agree with this comment by referee 3. In fact, FAD-linked presenilin mutants are only loss of function in the way that they show impaired processivity but not in the way stated by the "presenilin hypothesis" that the signaling cleavages are reduced. Since the term loss of function is commonly used in the field, we however felt to keep it in our manuscript. However, consistent with the term "shift of function" and that they are not loss of function in human brain, we have added a citation for the finding that γ -secretase activity of FAD mutant human patient brain does not show a loss of function (i.e. it displays rather normal initial ϵ -site cleavage) but persistently causes pathogenic A β ratios (Szaruga et al., 2015).

Cross-comments from referee 3:

I see the other comments are less enthusiastic... My own thought is that the following is reasonable to ask for the authors to address, but I still think they deserve a chance to respond.

-The authors propose that the RO7019009 GSM may specifically affect the A β 40 product line of the gamma secretase. However, this conclusion is not fully supported by the data. Kinetic investigation of the effects of this GSM on the gamma secretase proteolysis is needed in order to solve the intriguing findings.

-The authors discuss that the altered patterns could be linked to the changes in the strengths of the interactions between PS1 and APPC99. Importantly, the reduction in the strength of the gamma

secretase-APPC99 binding has been shown in two independent studies to play a key role in the promotion of the generation of longer A β species (Okochi et al., 2013, Szaruga et al., 2017). However, the referred studies indicate that pathogenic PS1 mutations affect substrate dissociation rather than its binding affinity. What is the view of the authors on this point? How can they put together the different findings?

I think this request is a tough one, one can get enzyme kinetics but I am skeptical about affinity measurements between CTF and gamma-secretase. I am not aware of anyone doing this affinity measurement, and if they have I am not sure that it would be meaningful..Indeed, I don't really know how one does such a measurement in a detergent containing solution. I think that is why the authors used this cross-linking approach.

-The crosslinking data presented in this study support the conclusion that FAD-linked mutations alter the interaction between the APPC99 substrate and the gamma secretase complex. The findings are intriguing but remain descriptive. The most important point is the lack of direct biochemical analysis (i.e. affinity constants) demonstrating that the differential crosslinking patterns actually reflect altered substrate binding as the authors conclude.

Response:

We would like to thank referee 3 for taking the time to cross-comment (*in italics*) on the concerns raised by referee 2 and specifying the comments which he thought are reasonable to address.

In his cross-comment, referee 3 particularly pointed out the difficulty to perform direct affinity measurements of APP CTFs with γ -secretase in line with our own thoughts on this most important point raised by referee 2. As already stated in our response to this point of referee 2 above and in agreement with the cross-comment, assays directly measuring APP CTF substrate binding affinities for γ -secretase are not established by any laboratory in the field including ours. In order to assess the binding affinities of γ -secretase to C99 a substantial effort would be needed to setup such a highly sophisticated assay and would most likely not bring substantially new information to support our principal findings and conclusions. We also want to stress again the point that the altered substrate interactions that we have observed are not meant to indicate that C99 binding affinities are changed.

To conclude, as described in the above-mentioned responses to referee 2, we added more discussion on the potential GSM mechanisms and additional data (**Figures EV3, EV4A and EV4B**) to further elucidate the differential effects of the GSM on the A β profiles of the mutants. We hope that we could satisfyingly address the points that referee 3 pointed out in his cross-comment.

Taken together, we believe that we have convincingly addressed all points raised by our referees and hope that our revised version will now become acceptable for publication in EMBO Reports.

Thank you for the submission of your revised manuscript. We have now received the enclosed reports from the referees that were asked to assess it. Referee 2 still has a few more suggestions that need to be addressed and incorporated before we can proceed with the official acceptance of your manuscript.

REFEREE REPORTS

Referee #2:

The authors have addressed satisfactorily the points that were raised by this reviewer. However, the revised manuscript is still unclear and even misleading when presenting and discussing the effects that FAD-linked mutations exert on the interactions between the APP substrate and the gamma secretase complex. Instead of 'binding' authors should refer to interactions or contacts between enzyme and substrate. Alternatively, results can be presented in terms of cross-linking levels.

Most importantly, the title must be adapted. As it is now it implies that FAD-mutations affect 'substrate binding'. This is clearly not supported by the presented data and in disagreement with previous reports.

In addition, some sentences need revision. Here some examples:

"Particularly, the V261F and R278I mutants displayed increased binding of K54 relative to the major interaction site V44 (Figure 2C). Consistent with our previous results [22], the L166P mutant showed shifts in binding at the T48 and L49 sites compared to WT (Figures 2A and B). Taken together, the mutants showed distinct and individual changes of C99 binding as compared to the respective WT and M292D controls."

" Differential effects on substrate binding in the presence of RO7019009 "

"Interestingly, as observed previously for the PS1 A246E and L166P mutants [22], binding alterations of residues T48 and L49,..."

Referee #3:

The authors have carefully addressed concerns raised by all reviewers in the manuscript.

Authors made the suggested changes.

Corresponding Author Name: Harald Steiner

Journal Submitted to: EMBOR

Manuscript Number: EMBOR-2019-47996